# Pharmacist-Driven Antibiotic Stewardship Program in Febrile Neutropenic Patients: A Single Site Prospective Study in Thailand

**DOI:** 10.3390/antibiotics10040456

**Published:** 2021-04-17

**Authors:** Kittiya Jantarathaneewat, Anucha Apisarnthanarak, Wasithep Limvorapitak, David J. Weber, Preecha Montakantikul

**Affiliations:** 1Department of Pharmacy, Faculty of Pharmacy, Mahidol University, Bangkok 10400, Thailand; kittiyaj@staff.tu.ac.th; 2Department of Pharmaceutical care, Faculty of Pharmacy, Thammasat University, Pathum Thani 12120, Thailand; 3Division of Infectious Diseases, Faculty of Medicine, Thammasat University, Pathum Thani 12120, Thailand; anapisarn@yahoo.com; 4Division of Hematology, Faculty of Medicine, Thammasat University, Pathum Thani 12120, Thailand; Wasithep@tu.ac.th; 5School of Global Public Health, University of North Carolina, Gillings, Chapel Hill, NC 27599-7400, USA; David.Weber@unchealth.unc.edu

**Keywords:** antibiotic stewardship, febrile neutropenia, appropriateness, pharmacist-driven, hematology oncologic patient

## Abstract

The antibiotic stewardship program (ASP) is a necessary part of febrile neutropenia (FN) treatment. Pharmacist-driven ASP is one of the meaningful approaches to improve the appropriateness of antibiotic usage. Our study aimed to determine role of the pharmacist in ASPs for FN patients. We prospectively studied at Thammasat University Hospital between August 2019 and April 2020. Our primary outcome was to compare the appropriate use of target antibiotics between the pharmacist-driven ASP group and the control group. The results showed 90 FN events in 66 patients. The choice of an appropriate antibiotic was significantly higher in the pharmacist-driven ASP group than the control group (88.9% vs. 51.1%, *p* < 0.001). Furthermore, there was greater appropriateness of the dosage regimen chosen as empirical therapy in the pharmacist-driven ASP group than in the control group (97.8% vs. 88.7%, *p* = 0.049) and proper duration of target antibiotics in documentation therapy (91.1% vs. 75.6%, *p* = 0.039). The multivariate analysis showed a pharmacist-driven ASP and infectious diseases consultation had a favorable impact on 30-day infectious diseases-related mortality in chemotherapy-induced FN patients (OR 0.058, 95%CI:0.005–0.655, *p* = 0.021). Our study demonstrated that pharmacist-driven ASPs could be a great opportunity to improve antibiotic appropriateness in FN patients.

## 1. Introduction

Febrile neutropenia (FN) is a life-threatening complication of cancer therapy which can increase morbidity and mortality [1]. Broad spectrum antimicrobial agent administration is an essential part of the treatment of febrile neutropenia to cover hospital-acquired pathogens. Pharmacokinetic alterations of several antibiotics (e.g., piperacillin/tazobactam) were found in febrile neutropenic patients [2,3]. Prescribing antibiotics with common dosage regimens might be inadequate for these patients. Furthermore, incorrect antibiotic dosing was found as the most common non-compliant antibiotic prescription practice in febrile neutropenic patients [4]. Antibiotic optimization would be a challenging method among febrile neutropenic patients. An antibiotic stewardship program (ASP) in immunocompromised patients is suggested by the Infectious Diseases Society of America (IDSA) 2010 guidelines [5]. Recent evidence supports that adherence to an ASP is associated with a lower mortality rate [6]. Although several studies have shown the effectiveness of ASP implementation in febrile neutropenic patients, there is limited evidence of the effectiveness of ASP implementation led by a pharmacist [7,8,9,10,11]. Pharmacist-driven ASPs have been reported to increase antibiotic appropriateness in several studies [12,13,14]. We believe this is the first study to demonstrate that a pharmacist-driven ASP can be beneficial among febrile neutropenic patients. Our study compared antibiotic appropriateness between a pharmacist-driven ASP and a control group.

## 2. Results

Ninety febrile neutropenic events occurred in 66 patients. The proportion of men in the control group was higher than the intervention group (57.8% vs. 35.6%, *p* = 0.035). The mean age of all patients was 51.6 ± 15.6 years. Most patients were diagnosed with cytotoxic chemotherapy-induced febrile neutropenia (74.4%) while twenty patients were identified as a febrile neutropenia during the period of initial hematologic abnormalities diagnosis (22.2%) and only three patients were diagnosed as a febrile neutropenia from other causes such as vitamin B12 deficiency, severe infection, and zidovudine-induced pancytopenia. The majority of our patients had hematologic malignancy (80%) and 8.9% had solid cancer. The Multinational Association for Supportive Care in Cancer (MASCC) risk index median score was 20 (interquartile range (IQR) 17–21). The median absolute neutrophil count was 153.9 cells/mm^3^ (IQR 19–520). Fifty-one percent of patients had a history of febrile neutropenia and 55.6% of patients had been exposed to antibiotics within the past 3 months. The median duration of neutropenia was 7 days. The frequency of infectious diseases consultation was similar in both groups. The baseline characteristics are displayed in Table 1.

The major causative organisms were Gram-negative bacteria (43.3%), followed by Gram-positive bacteria (13.3%) and fungi (3.3%). The most common causative Gram-negative bacteria were *Escherichia coli* (33.3%), *Klebsiella pneumoniae* (25.6%), and *Pseudomonas aeruginosa* (10.3%). Most Gram-negative bacteria exhibited multiple-drug resistance (MDR) (69%). More carbapenem-resistant Gram-negative bacteria were often found in the pharmacist-driven ASP group compared to the control group (8.9% vs. 2.2%, *p* = 0.167) while extended spectrum beta-lactamase (ESBL)-producing Gram-negative bacteria were lower than the control group (6.7% vs. 20%, *p* = 0.063). The most common causative Gram-positive bacteria were *Enterococci* spp. (41.7%), *Staphylococcus aureus* (33.3%), and *Corynebacterium* spp. (16.7%). Ampicillin-resistant *Enterococci* spp. was isolated from only in one patient and only one patient had methicillin-resistant *Staphylococcus aureus* (MRSA). Most of the causative organisms were isolated from blood, urine, or sputum (27.8%, 12.2%, and 8.9%, respectively). The most common sources of infection were primary bacteremia, urinary tract infection, and pneumonia (23.3%, 13.3%, and 10%, respectively). However, the causative organisms were not isolated in nearly half of patients.

Overall, antibiotic appropriateness in the pharmacist-driven ASP group was significantly higher than the control group (88.9% vs. 51.1%, *p* < 0.001) (Table 2). In providing empirical therapy, the pharmacist-driven AS*P* group was more appropriate than the control group (97.8% vs. 77.8%, *p* = 0.007). The appropriate dosage regimen in the pharmacist-driven ASP group was significantly higher than the control group (97.8% vs. 88.7%, *p* = 0.049) as well as appropriate antibiotic coverage (100% vs. 91.1%, *p* = 0.041), while appropriate indications were similar in both groups. When providing therapy for definitive infections, the overall appropriateness was greater in the pharmacist-driven ASP group than in the control group (88.9% vs. 64.4%, *p* = 0.004), as was the duration of therapy (91.1% vs. 75.6%, *p* = 0.039). For therapy if the source of infection was unknown, the overall appropriateness in the pharmacist-driven ASP group also significantly greater than the control group (90% vs. 54.4%, *p* = 0.011). Furthermore, the appropriateness of duration of therapy in the pharmacist-driven group was significantly greater than in the control group (93.2% vs. 75.6%, *p* = 0.022). However, the antibiotic appropriateness in cases of known causative pathogens were not significantly greater than the control group, but there was a trend of improved appropriateness in the pharmacist-driven ASP group. The total antibiotic duration between two groups were similar (*p* = 0.948) (Table 2). The compliance rate to the pharmacist suggestion was 93.8% in the pharmacist-driven ASP group. The most common pharmacist interventions were de-escalation (31.3%), adding additional antimicrobials (18.8%), and avoiding serious drug interaction (18.1%).

Overall, the antibiotic appropriateness in the pharmacist-driven ASP group was significantly higher than control group (88.9% vs. 51.1%, *p* < 0.001) (Table 2). In providing empirical therapy, the pharmacist-driven ASP group was more appropriate than the control group (97.8% vs. 77.8%, *p* = 0.007). The appropriate dosage regimen in the pharmacist-driven ASP group was significantly higher than the control group (97.8% vs. 88.7%, *p* = 0.049) as well as the appropriate antibiotic coverage (100% vs. 91.1%, *p* = 0.041) while the appropriate indications were similar in both groups. When providing therapy for definitive infections, the overall appropriateness was greater in the pharmacist-driven ASP group than in the control group (88.9% vs. 64.4%, *p* = 0.004) as was the duration of therapy (91.1% vs. 75.6%, *p* = 0.039). For therapy if the source of infection was unknown, the overall appropriateness in the pharmacist-driven ASP group was also significantly greater than the control group (90% vs. 54.4%, *p* = 0.011). Furthermore, the appropriateness of duration of therapy in the pharmacist-driven group was significantly greater than in control group (93.2% vs. 75.6%, *p* = 0.022). However, the antibiotic appropriateness in cases of known causative pathogens were not significantly greater than the control group, but there was a trend of improved appropriateness in the pharmacist-driven ASP group. The total antibiotic duration between two groups were similar (*p* = 0.948) (Table 2). The compliance rate to the pharmacist suggestion was 93.8% in the pharmacist-driven ASP group. The most common pharmacist interventions when compared with the control group were de-escalation (22.2% vs. 20%, *p* = 0.796), adding additional antimicrobials (17.8% vs. 8.9%, *p* = 0.215), and avoiding serious drug interaction (6.7% vs. 0%, *p* = 0.078). 

The 30-day infectious diseases-related mortality and length of stay were similar in both groups (Table 2). In univariate analysis, neither the pharmacist-driven ASP nor ID consultation showed a significant impact on 30-day infectious diseases-related mortality (*p* = 0.810 and 0.267, respectively). However, in multivariate analysis, the pharmacist-driven ASP group and infectious diseases consultation significantly reduced the 30-day infectious diseases mortality in patients with cytotoxic chemotherapy-induced febrile neutropenia (OR 0.058, 95% CI: 0.005–0.655, *p* = 0.021). A history of febrile neutropenia was associated with an increased 30-day infectious diseases mortality, as described in Table 3. The utilization rate of target antibiotics in the pharmacist-driven ASP group tended to be higher than control group (882 Defined Daily Dose (DDD)/1000 patient-day vs. 705.1 DDD/1000 patient-day). The trend of the overall target antibiotic seemed to be higher in both groups (Appendix A). The trend of ceftazidime, cefepime, and meropenem utilization was lower in the pharmacist-driven ASP group while piperacillin/tazobactam utilization was higher. In the control group, ceftazidime utilization tended to be decreased, but other target antibiotics’ utilization including cefepime, piperacillin/tazobactam, and meropenem were increased. Overall intravenous antibiotic utilization in the pharmacist-driven ASP group declined while amount of utilization in the control group increased.

## 3. Discussion

The pharmacist-driven ASP group interventions in febrile neutropenic patients showed a favorable effect on antibiotic appropriateness in our study. We found higher antibiotic appropriateness in the pharmacist intervention group than the control group (88.9% vs. 51.1%, *p* < 0.001). When providing empirical therapy, the pharmacist-driven ASP group was more appropriate than the control group, which was different from a previous study [15]. We believe that main reason for this discrepancy was that the previous study evaluated only the prescribed antibiotic appropriateness based on the hospital guidelines and described only the antibiotics indicated for febrile neutropenia, but did not assess the appropriateness of dosage regimens which was included in our assessment of the appropriateness of prescribed antibiotics [15,16]. Moreover, our ASP implementation provided daily review and feedback while the Madran et al. study implemented a hospital guideline and had only a weekly discussion with the ASP team [15]. Likely our study, which provided more frequent feedback, improved the primary physicians’ compliance, as noted in a recent study [10]. Furthermore, our results showed that the appropriate dosage regimens were more frequently found in the pharmacist-driven ASP group than the control group (97.8% vs. 88.7%, *p* = 0.049). Our finding was similar to previous studies that the appropriateness was 6.5-fold higher in the pharmacist intervention group then the control group [17]. On the other hand, the appropriateness of antibiotic indication in the pharmacist-driven ASP group resembled the control group, as described in Madran et al. study [15]. However, we also used current standard guidelines and all of our patients had a high risk of febrile neutropenia, similar to the previous study [15].

In documented infection evaluations, the pharmacist-driven ASP group had a greater appropriateness of prescribed antibiotics than the control group (88.9% vs. 64.4%, *p* = 0.004). Our result was similar to previous study in which more appropriateness was found in the intervention group [15]. Moreover, the appropriate duration of therapy was higher in the pharmacist-driven ASP group (*p* = 0.039). Our result was concordant with a previous study that pharmacist-driven ASPs could reduce the duration of antibiotic therapy [18]. However, the appropriateness of antibiotic indication was similar in both groups because our study divided the category of appropriateness into microbial susceptibilities and the penetration of antibiotics to the target site. If pathogens were identified and antibiotics susceptibilities were reported, it could help physicians to choose proper antibiotics. Since most pathogens in the control group were ESBL-producing organisms, this might affect antibiotic appropriateness because carbapenems are drugs of choice for ESBL-producing organisms, and choice of antibiotics was controlled by an infectious diseases physician [19]. In addition, the overall antibiotic appropriateness and proper duration of therapy in the pharmacist-driven ASP group were also greater than the control group when the source of infection was unknown (*p* = 0.039 and 0.066, respectively) (Appendix A). However, the total antibiotic duration between two groups did not differ. The reasons for prolonging antibiotic duration in the intervention group were fungal infection (e.g., invasive pulmonary aspergillosis and mucormycosis), superinfection with MDR organisms, and an uncontrolled source of infection. Although the result did not show any difference of antibiotic appropriateness in the case of known causative pathogens and source of infection between the two groups, the pharmacist-driven ASP group tended to use more proper antibiotics than the control group in terms of indication, dose and duration (*p* = 0.384, 0.833, and 0.872, respectively) (Appendix A).

Nevertheless, our study did not show a difference in the 30-day infectious diseases-related mortality between the two study groups. Our patients tended to have longer neutropenia durations and higher carbapenem resistance Enterobacteriaceae (CRE) infection rates in the intervention group which differed from a previous study [15]. As a result of rising CRE incidence in Thailand, our patients were more likely to be infected with CRE than reported in a previous study, which might have affected the mortality rate in our study [15,20,21]. However, other ASP studies in febrile neutropenic patients also showed no difference in mortality between two groups as well [10,22,23,24,25]. Based on our multivariate analysis, pharmacists should collaborate with other medical personnel such as infectious diseases physicians to improve the 30-day infectious diseases-related mortality in febrile neutropenic patients caused by cytotoxic chemotherapy. This study supports the IDSA Guidelines that ASP team should be made up of a multidisciplinary team to achieve successful ASP implementation [5]. On-site infectious diseases specialists, including an ID physician and pharmacist, can improve the ASP effectiveness in recent study [26]. Hence, a multidisciplinary team would be beneficial for ASP implementation in these specific populations where data are limited, such as febrile neutropenic patients. Notably, although most of the febrile neutropenic patients in our study were caused by cytotoxic chemotherapy, there are some patients caused by hematologic abnormalities during diagnosis in our study which was also mentioned in previous study [27]. Furthermore, our study did not find any difference in the length of stay in both groups, as has been noted in a previous study [10].

Although our target antibiotics utilization in the pharmacist-driven ASP group increased during the study period, which was similar to a previous study, it might have affected inappropriate prescriptions in the control group [10]. For instance, there were some antibiotics improperly used in empirical therapy in the control group such as ceftriaxone, which were not included in our target antibiotics, and antibiotics might have been prescribed at an improper low dose. Therefore, the DDD of target antibiotics in the control group might be lower than expected. Moreover, we implemented a high dose of target antibiotics according to previous pharmacokinetic studies this might have contributed to a higher DDD of target antibiotic in the pharmacist-driven ASP group [2,3]. Besides, the overall intravenous antibiotics in the pharmacist-driven ASP group demonstrated a lower trend than the control group.

Our study had several limitations. First, ward physician rotation could have affected the result. However, the result of this study also showed that the pharmacist intervention group had more appropriateness than the control group. Second, the study was implemented only in medical wards since Thammasat University Hospital did not have a hematology–oncology ward during the study period and we could not fully perform interventions in the other wards such as the emergency department and intensive care unit. Ideally, the ASP implementations in febrile neutropenic patients should be carried out in all wards. Third, we calculated our sample size to demonstrate antibiotic appropriateness rather than 30-day infectious diseases-related mortality. A larger sample size is needed to assess the effect of a pharmacist-driven ASP on 30-day infectious diseases-related mortality. Fourth, we could not evaluate the effect of pharmacist-driven ASPs on antibiotic resistance since the study site did not have an isolation ward for febrile neutropenia patients with multidrug-resistant pathogens. Thus, the acquisition of antibiotic resistance organisms from other patients might have affected our results. Finally, the role of pharmacists in Thailand may be different from western countries. Pharmacists cannot change the antibiotic dosage regimen or discontinue antibiotics by themselves; a physician’s signature is needed. Thus, pharmacist cooperation with a physician was also an important aspect to implement a successful pharmacist-driven ASP in Thailand.

In conclusion, our study showed that a pharmacist-driven ASP in febrile neutropenic patients could improve the antibiotic appropriateness in both empirical and documentation therapy. However, 30-day infectious diseases-related mortality and the length of stay were not different between the groups. Although the target antibiotic utilization in the intervention group increased, we found a reduction in the total antibiotic utilization in the pharmacist-driven ASP group.

## 4. Materials and Methods 

This prospective study was conducted at Thammasat University Hospital in Thailand, a tertiary care and teaching hospital, between 1 August 2019 and 30 April 2020. Two medical wards were pre-designated as the pharmacist-driven ASP group and two other similar medical wards were pre-designated as the control group. Febrile neutropenia in our study was defined as fever (single temperature equivalent to ≥38.3 °C orally or equivalent to ≥38.0 °C orally over a 1 h period) with neutropenic condition (patient who had ≤500 neutrophils per microliter or ≤500 neutrophils per microliter and a predicted declined to ≤500 neutrophils per microliter over the next 48 h). High risk of febrile neutropenia was identified by the MASCC risk index score less than 21 [28]. Inclusion criteria included adult patients (i.e., age >18 years); patient diagnosed with febrile neutropenia; and patient received antibiotics for treatment of febrile neutropenia. Exclusion criteria included the receipt of antibiotics for febrile neutropenia <24 h, pregnancy, or lactation. This study was approved by the human research ethics committee, Thammasat University (protocol no. MTU-EC-OO-0-078/62).

Our ASP team consisted of an infectious diseases-trained clinical pharmacist, infectious diseases physicians and hematologists. We developed TUH’s recommended antibiotic and dosage regimen for empirical therapy in febrile neutropenia, which, adapted from the IDSA 2010 and National Comprehensive Cancer Network (NCCN) 2020 guidelines and distributed to primary physicians prior to the pharmacist-driven ASP, was implemented in two medical wards groups [5,28]. In the intervention group, a clinical pharmacist performed the daily prospective audit and feedback to the primary physician. The pharmacist suggested a suitable antibiotic for each patient, calculated an appropriate dose and recommended the treatment duration for both empirical therapy and documented infection. The antibiotic appropriateness and antibiotic utilization in the intervention group was reported monthly by the clinical pharmacist. Medical personnel practicing in the intervention group were provided education via lectures and posters by the clinical pharmacists during monthly ward conferences. No ASP interventions were performed in the control group. The criteria to evaluate antibiotic appropriateness was adapted from previous studies (Appendix A) [29,30,31]. In empirical therapy evaluations, a clinical pharmacist evaluated an appropriateness of indications, antibiotic coverage, and dosage regimen of the antibiotics. Therapeutic evaluations for documented infection were divided into 2 groups—unknown source of infection and known causative pathogens and source of infection. Both groups were also evaluated for antibiotic indication, dosage regimen, and duration of antibiotic therapy by pharmacist.

The primary outcome of this study was to compare antibiotic appropriateness between pharmacist-ASP driven group and the control group. Secondary outcomes were to compare antibiotic utilization, patient length of stay, 30-day infectious diseases-related mortality between the intervention and control groups. Target antibiotics in this study were ceftazidime, cefepime, piperacillin/tazobactam, meropenem and imipenem which are recommended as an empirical therapy for febrile neutropenia in current guidelines [28,32,33]. All intravenous antibiotics classes commonly used in these patients were evaluated in this study.

To have an 80% power and 95% confidence interval, the minimum sample size required in each arm, calculated based on a previous study, was 33 subjects [15]. Each outcome was defined as a febrile neutropenic event. All statistical analyses were performed using STATA version 16 (College Station, TX). Chi-square test (two-tailed) was used to compare proportion for categorical variables while t-test was used to compare means for continuous variables. Antibiotic utilization was reported as the defined daily dose per 1000 patient-days. The trend of antibiotic utilization was analyzed by linear regression and reported as the coefficient and *p*-value. Univariate and multivariate analyses of variables influencing on 30-day infectious diseases-related mortality were performed. All comparisons were 2-sided and a *p* value < 0.05 was consider statistically significant.

## Figures and Tables

**Table 1 antibiotics-10-00456-t001:** Baseline characteristics.

Baseline Characteristic	Total (90 FN Episodes), No. (%)	Intervention (45 FN Episodes), No. (%)	Control (45 FN Episodes), No. (%)	*p* Value
Age, mean years ± SD	51.6 ± 15.6	15.6 ± 14.6	52.0 ± 16.7	0.894
Male	42 (46.7)	16 (35.6)	26 (57.8)	0.035
Weight, mean kg ± SD	57.76 ± 1.50	58.94 ± 1.94	60.57 ± 2.30	0.590
Cause of febrile neutropenia				
Cytotoxic chemotherapy	67 (74.4)	34 (75.6)	33 (73.3)	1.000
During period of initial hematologic abnormalities diagnosis	20 (22.2)	10 (22.2)	10 (22.2)	1.000
Other causes *^a^*	3 (3.33)	1 (2.22)	2 (4.44)	1.000
Active hematologic cancer	72 (80)	34 (75.6)	38 (84.4)	0.496
Active solid cancer	8 (8.9)	4 (8.9)	4 (8.9)	0.496
MASCC score, median (IQR)	20 (17–21)	19 (13–21)	21 (19–21)	0.129
High risk of febrile neutropenia (MASCC < 21)	45 (50)	25 (55.6)	20 (44.4)	0.292
Absolute neutrophil count, median cells/mm^3^ (IQR)	153.9 (19–520)	184 (40–645)	77 (13–368)	0.198
Had history of febrile neutropenia	46 (51.1)	20 (44.4)	26 (57.8)	0.206
Recent exposed to antibiotic within past 3 months	50 (55.6)	25 (55.6)	25 (55.6)	1.000
Neutropenia duration, median days (IQR)	7 (4–14)	8 (4–14)	6 (4–10)	0.435
Infectious diseases specialist consultation	50 (55.6)	27 (60)	23 (51.1)	0.396
Time to administer antibiotic, median hours (IQR)	1 (0–4)	1.5 (0–4)	1 (0–4)	0.497
Causative organism identified	49 (54.4)	26 (57.8)	23 (51.1)	0.525
Gram-positive bacteria	12 (13.3)	7 (15.6)	5 (11.1)	0.774
Gram-negative bacteria	39 (43.3)	20 (44.4)	19 (42.2)	0.761
ESBL-producing organisms	12 (13.3)	3 (6.7)	9 (20)	0.118
Carbapenem resistance organisms	5 (5.6)	4 (8.9)	1 (2.2)	0.361

ESBL, extended spectrum beta-lactamase; FN, febrile neutropenia; IQR, interquartile range; MASCC, Multinational Association for Supportive Care in Cancer risk index score; SD, standard deviation. *^a^* Other causes of febrile neutropenia were from vitamin B12 deficiency, zidovudine-induce pancytopenia and severe infection.

**Table 2 antibiotics-10-00456-t002:** Study outcomes.

Outcomes	Intervention (45 FN Episodes), No. (%)	Control (45 FN Episodes), No. (%)	*p* Value
Overall appropriateness	40 (88.9)	23 (51.1)	<0.001
Step 1 Empirical therapy	44 (97.8)	35 (77.8)	0.007
Appropriate Indication	45 (100)	45 (100)	-
Appropriate coverage	45 (100)	41 (91.1)	0.041
Appropriate dosage regimen	44 (97.8)	39 (88.7)	0.049
Step 2 Documentation therapy	40 (88.9) *^b^*	29 (64.4)	0.004
Appropriate indication	43 (95.6)	41 (91.1)	0.361
Appropriate dosage regimen	44 (97.8)	43 (93.3)	0.242
Appropriate duration	41 (91.1)	34 (75.6)	0.039
Length of stay, median days (IQR)	28 (19–42)	23 (16–35)	0.689
30-day infectious diseases related mortality	6 (13.6)	5 (11.1)	1.000
Total antibiotic duration, median days (IQR)	14 (10–23)	15 (10–21)	0.948
antibiotic duration in de-escalation	21 (14–28)	17.5 (15.5–29.5)	0.666
antibiotic duration in escalation	19 (13–34.5)	15 (11–25.5)	0.309

FN, febrile neutropenia; IQR, interquartile range. *^b^* Total 44 FN episodes since one death occurred before culture was reported.

**Table 3 antibiotics-10-00456-t003:** Multivariate analysis of 30-day infectious diseases-related mortality.

Variables	Univariate Analysis			Multivariate Analysis		
	OR	95%CI	*p* Value	OR	95%CI	*p* Value
Pharmacist-driven ASP group and infectious diseases consultation in chemotherapy-induced febrile neutropenic patient	0.184	0.037–0.911	0.038	0.058	0.005–0.655	0.021
Male	0.653	0.176–2.419	0.524	0.744	0.133–4.148	0.736
High risk of febrile neutropenia	5.426	1.098–26.829	0.038	5.155	0.762–34.890	0.093
Had history of febrile neutropenia	5.143	1.040–25.420	0.045	9.380	1.311–67.100	0.026
Carbapenem resistance organisms	8.111	1.015–64.839	0.048	18.771	0.560–628.848	0.102
ESBL producing Gram negative bacteria	2.75	0.614–12.307	0.186	7.417	0.787–69.906	0.080

ASP, antibiotic stewardship program; CI, confidence interval; ESBL, extended spectrum beta-lactamase; MASCC, Multinational Association for Supportive Care in Cancer risk index score; OR, odds ratio.

## Data Availability

The data that support the findings of this study are available on request from the corresponding author. The data are not publicly available due to privacy or ethical restrictions.

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
