# Peer review of "Pharmacist-Driven Antibiotic Stewardship Program in Febrile Neutropenic Patients: A Single Site Prospective Study in Thailand"

_antibiotics, 2021, doi:10.3390/antibiotics10040456_

Round 1
Reviewer 1 Report
This well designed study describes the impact of a pharmacist-driven ASP on the appropriate use of antibiotics in patients with febrile neutropenia. Your results are impressive, and are presented clearly, and your conclusions are supported by the results.
My one concern is your sample size statement on Page 8 ("Minimum sample size required in each arm, calculated based on a previous study, were 33 subjects"). The Madran et al study is similar in design but has a different primary outcome. Using "appropriate adding or changing antimicrobials" (81% in ASP cohort vs. 53% without ASP) from Madran et al, I calculate a required sample size of 144 subjects.
I feel that your paper will be greatly enhanced by describing your sample size (power) calculation, specifically your assumptions based on Madran et al.
Reviewer 2 Report
Dear sir/madam,
This is a very interesting study aiming to determine role of pharmacist in ASP for 18 FN patients. The authors showed that pharmacist-driven ASP in febrile neutropenic patients could improve antibiotic appropriateness in both empirical and documentation therapy. However, 30-day infectious diseases-related mortality and length of stay were not different between the groups. Although target antibiotic utilization in the intervention group increased, they found a reduction in total antibiotic utilization in the pharmacist-driven ASP group.
Overall, antibiotic appropriateness in the pharmacist-driven ASP group was significantly higher than in the control group. Appropriate dosage regimen in the pharmacist-driven ASP group was significantly higher than in the control group as well as appropriate antibiotic coverage while appropriate indications was similar in both groups. Furthermore, appropriateness of duration of therapy in the pharmacist-driven group was significantly greater than in control group. Overall intravenous antibiotic utilization in the pharmacist-driven ASP group declined while amount of utilization in the control group increased.
The most common pharmacist interventions were de-escalation, adding additional antimicrobials, and avoiding serious drug interaction. Please mention whether these differences were statistically significant.
More carbapenem resistance Gram negative bacteria were often found in the pharmacist-driven ASP group compared to control group while extended spectrum beta-lactamase (ESBL) producing Gram negative bacteria were lower than the control group. I suggest that these parameters are included in the multivariable regression analysis model.
Page 7 prescribes: to be replaced by prescribed
Page 6 which also mentioned in previous study: to be replaced by which was also mentioned in a previous study
Page 7 TUH: do not use abbreviation
I suggest that this paper is suitable for publication in antibiotics after minor revisions.
Yours sincerely
